# Synthesis and Characterization of Nano-Sized 4-Aminosalicylic Acid–Sulfamethazine Cocrystals

**DOI:** 10.3390/pharmaceutics13020277

**Published:** 2021-02-19

**Authors:** Ala’ Salem, Anna Takácsi-Nagy, Sándor Nagy, Alexandra Hagymási, Fruzsina Gősi, Barbara Vörös-Horváth, Tomislav Balić, Szilárd Pál, Aleksandar Széchenyi

**Affiliations:** 1Institute of Pharmaceutical Technology and Biopharmacy, University of Pécs, 7622 Pécs, Hungary; ala.salem@aok.pte.hu (A.S.); anna.takacsi-nagy.aman@aok.pte.hu (A.T.-N.); sandor.nagy@aok.pte.hu (S.N.); jim2qq@tr.pte.hu (A.H.); fruzsina.gosi@aok.pte.hu (F.G.); barbara.horvath@aok.pte.hu (B.V.-H.); 2Department of Chemistry, Josip Juraj Strossmayer University of Osijek, 31000 Osijek, Croatia; tombalic@kemija.unios.hr

**Keywords:** nano-drugs, 4-aminosalicylic acid, sulfamethazine, cocrystals, high-pressure homogenization, high-power ultrasound

## Abstract

Drug–drug cocrystals are formulated to produce combined medication, not just to modulate active pharmaceutical ingredient (API) properties. Nano-crystals adjust the pharmacokinetic properties and enhance the dissolution of APIs. Nano-cocrystals seem to enhance API properties by combining the benefits of both technologies. Despite the promising opportunities of nano-sized cocrystals, the research at the interface of nano-technology and cocrystals has, however, been described to be in its infancy. In this study, high-pressure homogenization (HPH) and high-power ultrasound were used to prepare nano-sized cocrystals of 4-aminosalysilic acid and sulfamethazine in order to establish differences between the two methods in terms of cocrystal size, morphology, polymorphic form, and dissolution rate enhancement. It was found that both methods resulted in the formation of form I cocrystals with a high degree of crystallinity. HPH yielded nano-sized cocrystals, while those prepared by high-power ultrasound were in the micro-size range. Furthermore, HPH produced smaller-size cocrystals with a narrow size distribution when a higher pressure was used. Cocrystals appeared to be needle-like when prepared by HPH compared to those prepared by high-power ultrasound, which had a different morphology. The highest dissolution enhancement was observed in cocrystals prepared by HPH; however, both micro- and nano-sized cocrystals enhanced the dissolution of sulfamethazine.

## 1. Introduction

Pharmaceutical cocrystals have been utilized to enhance the solubility and bioavailability [1], among other physicochemical properties, of active pharmaceutical ingredients (APIs). Aiming to develop pharmaceutical combination therapy or fixed-dose medication [2], pharmaceutical drug–drug cocrystals, where the coformer is also an API, offer a convenient approach when a specific disease is treated by a combination of APIs [3]. Combination therapy has gained increasing popularity due to several advantages such as improved therapeutic effect, reduced prescription numbers, lower cost of administration, and patient compliance [4]. Therefore, a main aim of formulating drug–drug cocrystals is not only the modulation of API physicochemical properties but the possibility of producing a combined medication as well, resulting in a reduced pill burden and minimizing medication errors. The first drug–drug cocrystal product was approved in 2015 by the United States Food and Drug Administration [5].

Tuberculosis (TB), an infectious disease usually caused by Mycobacterium tuberculosis [6], remains a main cause of mortality globally [7]. In 2016, TB was estimated to have infected 10.4 million [8], while causing 1.7 million deaths, making it the deadliest infection worldwide [9,10]. The treatment of TB necessitates the use of multiple medications for long periods of time. The standard regimen consists of isoniazid, rifampicin, pyrazinamide, and ethambutol for two months, followed by isoniazid and rifampicin for four additional months [11]. However, control of multidrug-resistant TB (MDR-TB) is failing [12,13], and treatment outcomes remain poor [14], while treatment has advanced very little [6]. Prevention through vaccination with the only available vaccine, bacillus Calmette–Guérin (BCG), can only decrease the risk of infection by 20% in vaccinated children and decreases the risk of turning the infection into active TB by 60% [15].

Patients failing to respond to multiple first-line TB medications have to use expensive second-line TB drugs, which include aminoglycosides, polypeptides, fluoroquinolones, thioamides, cycloserine, and 4-aminosalicylic acid (4-ASA) [16]. Many of these antibiotics were developed decades ago and suffer from toxic side effects, administration difficulty, and poor activity against Mycobacterium tuberculosis [6]. It is recommended that MDR-TB be managed with at least four effective antibiotics for 18 to 24 months [11,17]. Meanwhile, trials are set to find shorter and less debilitating regimens for MDR-TB [14,18].

Cocrystals have been utilized to enhance classic fixed-dose combinations of TB antibiotics. Isoniazid–caffeine/vanillic acid cocrystals were reported to have greater stability compared to the classical drug combination [19]. Cocrystals of 4-ASA with isonicotinamide were shown to have improved solubility and stability [20]. Moreover, cocrystals of 4-ASA with isoniazid were also reported to be more stable thermally [21]. The combination of 4-ASA and sulfamethazine (SMT) in cocrystals was proposed to exploit an anti-TB synergistic effect [22].

Particle size reduction can be aimed at API dissolution rate improvement [23]. Nano-sized crystals have gained substantial attention [24]. Generally, nano-crystals adjust the pharmacokinetic properties and enhance the penetration and distribution of APIs [25]. Nano-crystals are used to enhance the bioavailability of APIs by increasing dissolution velocity and saturation solubility [26,27]. Nano-crystals offer an opportunity to deliver poorly water soluble drugs, as they are a carrier-free colloidal system in the nano-sized range [28]. Therefore, the use of nano-crystals in pharmaceutical formulations has been increasing, as several techniques for the preparation of nano-crystals are routinely used in the pharmaceutical industry and a wide range of products are in clinical trials and even on the market [29]. With the rapid development of cocrystal and nano-crystal technology, nano-cocrystals seem to enhance API properties by combining the benefits of both nano- and cocrystals. One example was reported by Karunatilaka et al. [30], who showed that an engineered nano-sized cocrystal of trans-1,2-bis(4-pyridyl)ethylene and 5-cyanoresorcinol had unique chemical and mechanical properties. Furthermore, the use of cocrystals can improve API properties like hygroscopicity, compressibility, and friability [29]. A synergistic effect was achieved when nano-technology and cocrystallization were combined. Furosemide–caffeine nano-cocrystals were reported to have a dissolved concentration that was more than three times that of furosemide nanocrystals. Furthermore, the nano-cocrystals had a higher dissolution rate than unmilled cocrystals [31]. Similarly, baicalein–nicotinamide nano-cocrystals showed enhanced dissolution compared to the cocrystals and baicalein nanocrystals [32]. Phenazopyridine–phthalimide nano-cocrystals were also found to significantly enhance the release rate of phenazopyridine [33]. By knowing the benefits of these strategies, they can be used to further enhance fixed-dose combinations, as the number of coformers, and the field of their properties’ enhancement, is limited. Application of both strategies could be used for further dissolution rate enhancement of poorly water soluble APIs if one of the strategies does not enhance it to the appropriate rate [34]. The potential for nano-cocrystals, after production challenges are overcome, includes a prolonged half-life and localized drug delivery when incorporated into complex delivery systems [35].

High-pressure homogenization (HPH) is a top-down technology widely employed to prepare nano-crystals from micronized suspensions. The resulting nano-suspension can be administered orally. The two applied homogenization principles are piston-gap homogenization and micro-fluidization. Piston-gap HPH nanosuspension production can be carried out in water, water mixtures, and water-free media. The homogenizer power density, homogenization cycles, and temperature all influence the nano-crystal size [36]. HPH involves high energy input and is rather inefficient and time consuming [37]. Nevertheless, the advantages of HPH include scale-up production feasibility, avoiding the use of harsh solvents, and a sterilizing effect [36,38]. Moreover, drug nano-suspensions are found to exhibit long-term physical stability, while dried powder can be formulated into oral drug dosage forms [36]. The use of HPH as a novel approach to prepare cocrystals was reported by Fernández-Ronco et al. [39]. Furthermore, the effect of pressure and homogenization passes was investigated previously for pharmaceutical solids [40].

Ultrasound-assisted cocrystallization is an advantageous method for screening cocrystals. Cocrystallization can be carried out in combination with slurry or solution cocrystallization [41]. Ultrasound is defined as “mechanical sound waves in the frequency range of 20 kHz to several GHz.” The lower 20 kHz–5 MHz frequency region is power ultrasound, where more acoustic energy is generated to induce cavitation in the liquid [42]. The solubility of the cocrystal components was found to affect both ultrasound- and microwave-assisted cocrystallization products [43]. Ultrasound-assisted cocrystallization is also influenced by the choice of solvent and stoichiometric ratios of the API and coformer [44]. Sono-crystallization, the application of ultrasound in crystallization, can be optimized in terms of ultrasonic frequency, power, operating conditions, and duration. Moreover, ultrasound-assisted crystallization can impact the particle shape and size and the polymorphic form [42].

Despite the promising opportunities of nano-sized cocrystals, research at the interface of nano-technology and cocrystals has been described to be in its infancy [35,45]. In this regard, our aim in this study was to prepare and characterize nano-sized multi-drug cocrystals and also to establish the effect of homogenization and high-power ultrasound parameters on size distribution, dissolution rate, thermal stability, degree of crystallinity, and polymorphic form produced. For the purpose of examining the effect of the method parameters, different pressures were used to perform HPH, operating with different cycle runs, while high-power ultrasound cocrystallization was carried out using different amplitudes and different process durations.

## 2. Materials and Methods

### 2.1. Materials

4-Aminosalicylic acid (4-ASA) (98%) was purchased from Sigma Aldrich (Steinheim, Germany), sulfamethazine (SMT) (99%) from Acros Organics (Fair Lawn, NJ, USA), polyvinyl alcohol from Hungaropharma Zrt. (Budapest, Hungary), and ethanol (>99.9%) from VWR Chemicals (Radnor, PA, USA). Freshly prepared distilled water was used. All materials were used without further purification.

### 2.2. Preparation of Nano-Sized Cocrystals by High-Pressure Homogenization

Nano-sized cocrystallization was performed using HPH. Briefly, a 2% *w/w* water suspension of a 1:1 molar ratio of APIs (Figure 1), containing 0.4% polyvinyl alcohol as a stabilizer to enhance the dispensability of the solid phase and to prevent crystal regrowth, was homogenized using a lab-scale high-shear dispersing emulsifier (IKA, Staufen, Germany) at 17,500 rpm for 10 min. The water-based suspension was passed through the HPH equipment (Invensys APV2000, APV systems, Albertslund, Denmark) at room temperature. Three different pressures (300, 600, and 900 bar) and homogenization cycle numbers (*c*: 1, 3, and 5) were used for comparison. Filtered samples were air-dried before further drying in a desecrator at room temperature (JEOL, Tokyo, Japan) before further use.

### 2.3. Preparation of Cocrystals by High-Energy Ultrasound

Sono-crystallization was performed using high-power ultrasound (Sonopulse HD2200.2, Bandelin, Berlin, Germany). Briefly, a 2% *w*/*w* water-based suspension of a 1:1 molar ratio of APIs, containing 0.4% polyvinyl alcohol as a stabilizer to enhance the dispensability of the solid phase, was placed in a beaker, and a TT 13-mm-diameter sonotrode tip with a VS70 extension attachment was used to deliver ultrasonic waves; 50%, 70%, and 90% of a 20 kHz amplitude was used for 10 min, 70% amplitude was further used for 20 and 30 min, and pulses were set at 5 s separated by a gap of 1 s. The samples were stored, as described in previous section.

### 2.4. Preparation of Cocrystals by Fast Solvent Evaporation and Control

For comparison, non-micronized control cocrystals were prepared by fast solvent evaporation, as previously described [46]. Briefly, a 1:1 molar ratio of 4-ASA and SMT was dissolved in ethanol. After complete dissolution, the solution was transferred to a round-bottom flask and attached to a rotatory evaporator (Heidolph Laborota 4000, Heidolph Instruments, Schwabach, Germany) equipped with a vacuum pump at a rotation speed of 90 rpm. Fast solvent evaporation was carried out at room temperature (23 ± 1 °C). Samples were placed in a desiccator (JEOL, Japan) before further characterization. A control for morphology and size change was prepared using the same water suspension as that used in the preparation of cocrystals by HPH and high-energy ultrasound, containing the same ratio of the APIs and the stabilizer, without further processing.

### 2.5. Solid-State Characterization

#### 2.5.1. Transmission Electron Microscopy (TEM)

Diluted samples, from the suspensions produced by HPH and high-energy ultrasound, were mounted on copper 400 square mesh sample holders coated with EMR carbon support film. Microscopy examinations were performed using TEM (JEOL-1400 TEM, JEOL, Tokyo, Japan). TEM images were also used for particle size analysis, and an average size was measured manually from at least 30 crystals using ToupView software version 4.10 (ToupTek, Zhejiang, China). The mean particle width was used to represent the cocrystal size, and the length of needle-shaped cocrystals was also reported.

#### 2.5.2. Powder X-ray Diffraction

The dried powders were examined by powder X-ray diffraction (PXRD) using a Rigaku MiniFlex 600 X-ray diffractometer (Rigaku, Tokyo, Japan), Cu-Kα1 radiation, and Smart lab/MiniFlex guidance software version 2.0.2.1 (Rigaku, Tokyo, Japan). Gently pressed powders, after grinding, were placed in aluminum holders and analyzed at room temperature. Data were collected at a scanning rate of 3°/min, in the 2*θ* range of 5°–40°.

#### 2.5.3. Thermal Analysis

Differential scanning calorimetry (DSC) and thermogravimetric analysis (TGA) were performed using a Mettler Toledo DSC 821e instrument and STARe Evaluation software version 1.1 (Mettler Toledo, OH, USA). We sealed 18–20 mg of the powder samples in aluminum pans, which were then pierced to provide vent holes, and heated at a rate of 10 °C/min in the temperature range of 40–220 °C under a nitrogen purge. Temperatures reported refer to the onset of melting.

#### 2.5.4. Dynamic Light Scattering (DLS)

Suspensions, produced by HPH and high-energy ultrasound, before filtering, were further diluted, and the particle size distribution was examined by dynamic light scattering (Nano-S Nanosizer, Malvern Instruments, Malvern, UK) using Zetasizer software version 7.11 (Malvern Panalytisal, Malvern, UK).

### 2.6. Dissolution Study

Dissolution apparatus (ERWEKA DT 700, Erweka, Langen, Germany) was utilized to conduct the dissolution study of the different cocrystals. Initially, weighted cocrystal powders were placed in the dissolution apparatus containing 900 mL of water as a dissolution medium, equilibrated at 37.5 ± 1 °C with a rotation speed set at 150 rpm. Then, 5 mL samples were drawn at specific time intervals with replacement, filtered by a 0.2 μm PTFE membrane filter, and then analyzed by a UV–VIS spectrophotometer (Jasco V-670) at 244 nm and 299 nm. The multi-wavelength linear regression method was used to account for absorbance overlapping [47]. Calibration curves and further details of the dissolution study can be found in Appendix A.

### 2.7. Statistical Analysis

Statistical analysis was conducted by the one-tailed paired-sample *t*-test using IBM SPSS software version 25 (IBM, NY, USA) and Microsoft Excel. A *p*-value of ≤0.05 was considered statistically significant.

## 3. Results and Discussion

### 3.1. Solid-State Characterization

The morphology of the cocrystals prepared by both methods was examined by TEM. Cocrystals prepared by HPH were needle-shaped, as observed by TEM (Figure 2a–c). Cocrystals prepared by high-power ultrasound, on the other hand, appeared to be tabular-shaped polyhedrons (Figure 2d–h). These appeared more similar to the shape of non-micronized form I cocrystals previously reported [46]. Crystal shape modification by ultrasound has been previously reported for potash aluminum crystals, which were found to be polyhedral instead of octahedral. This crystal shape modification was attributed to the growth rate increase of the new faces. However, a conclusion has not been reached as to the exact underlying mechanism [48]. Similarly, using ultrasound-assisted crystallization, Hazra et al. [49] reported a gradual change in the alpha calcium sulfate hemihydrate crystal morphology from long rods to hexagonal plates and then to plate-like with increasing surfactant concentration.

The polymorphic form of the produced cocrystals was determined by PXRD, as it has been reported that there are three forms of 4-ASA–SMT cocrystals [46]. PXRD patterns of cocrystals prepared by HPH and high-power ultrasound exhibited characteristic peaks of the reported cocrystal form I (Figure 3). This form was reported to be thermodynamically more stable than form II in aqueous solution [22]. Pattern indexing by EXPO software, using Mc Maille indexing, identified the cocrystals as Triclinic, P-1. This is also in agreement with the published form I of the cocrystals [50].

The degree of crystallinity was estimated by PXRD pattern analysis using representative peak integrals and intensities at 10.4° ± 0.1° for the 4-ASA–SMR cocrystal polymorph I [22]. Non-micronized cocrystals were considered to be of pure phase I. The degree of crystallinity was found to decrease with increasing HPH pressure (40.8%, 61.4%, and 84.5% for 900, 600, and 300 bar, respectively). Cocrystals prepared by high-power ultrasound had comparable crystallinity regardless of amplitude and time, ranging from 72.3% for 70% 30 min to 79.7% for 50% 10 min. Induction time and metastable zone width were found to be shortened with higher ultrasonic power. Moreover, ultrasound crystallization was reported to facilitate growth of the stable crystal form [51]. However, HPH cocrystals appeared crystalline in TEM images (Figure 2). PXRD can be used to assess the degree of crystallinity, as peak broadening or amorphous halos can indicate amorphous phases; however, these are not necessarily detectable by PXRD [52]. Moreover, PXRD peak broadening due to size reduction [53] affects proper estimation of the degree of crystallinity.

DSC thermograms of the cocrystals (Figure 4 and Appendix A) show endothermic peaks consistent with form I melting onset at 162–164.7 °C [22]. Cocrystals prepared by HPH had melting endotherms around 161 °C, while those prepared by high-power ultrasound had higher endotherms around 172 °C. These differences in melting point can be explained by the differences in size and shape and are consistent with the reported decrease in melting entropy and enthalpy with particle size decrease observed in other crystals [54]. Furthermore, the melting point of drugs has been reported to dramatically drop as the size reaches the nano-scale [55]. Narrower peaks were detected for the cocrystals produced by HPH compared to those prepared by high-power ultrasound. This is in agreement with the TEM images (Figure 2) and narrower PXRD peaks (Figure 3). The melting of cocrystals was accompanied by thermal decomposition. Thermal decomposition of pure 4-ASA occurs at 145 °C; the decomposition at this temperature does not occur in any cocrystal products (Appendix A).

Particle size distribution was determined by DLS and analysis of TEM images. DLS revealed that higher homogenization pressure in HPH resulted in a smaller particle size. Cocrystals prepared at 900 bar and with 5 homogenization cycles were found to be 190 ± 55.2 nm using DLS. A similar particle size was reported for baicalein–nicotinamide cocrystals prepared by HPH using Poloxamer 188 as a stabilizer [32]. Unlike cocrystals prepared at 900 bar, the cocrystals prepared at 300 and 600 bar had multimodal size distribution, as three peak sizes were seen on DLS (459 nm, 2.3 and 6.4 μm; 295 nm, 1.4 and 5.6 μm, respectively) (Appendix A). The multimodal size distribution is a result of the needle-like crystal shape and agglomeration of needle-shaped crystals, and it is unsuitable for exact size distribution determination by DLS. The precise size distribution was determined from TEM image analysis (Table 1). The sedimentation observed in the suspensions prepared by high-power ultrasound made establishing an accurate size from DLS difficult (Appendix A); therefore, data from TEM image analysis were used. Cocrystals prepared by high-power ultrasound using the different amplitudes had statistically significant smaller sizes than the control (*p* ≤ 0.001 for all samples, except 50% 10 min: *p* = 0.048). Cocrystals prepared using higher ultrasound amplitude percentages were found to have smaller sizes with a narrower size distribution: 4.47 ± 3.72 μm for 50% amplitude compared to 2.59 ± 2.31 μm for 90% amplitude. Increasing the duration from 20 to 30 min also had a similar effect; however, this was not statistically significant. This is in agreement with the published data, as Amara et al. [48] reported a decrease in size with increasing ultrasound power and a narrower size distribution with longer duration. The water-based suspensions used in dynamic light scattering might not be appropriate for this method to accurately represent the average particle size, as the particle shape, together with aggregates and sedimentation observed in the samples, affects the method’s accuracy [56].

### 3.2. Dissolution

The rate of dissolution of SMT was statistically higher from all cocrystals compared to the control, which exhibited a dissolution profile between the two APIs. Likewise, the use of 4-ASA in this cocrystal pair was regarded as the coformer, rationalized to increase the dissolution of the poorly-water-soluble SMT [57]. Dissolution from the nano-sized cocrystals prepared using HPH was higher than from those prepared by high-power ultrasound. This can be expected, as dissolution from nano-sized cocrystals is higher than that from micro-size cocrystals [58]. Cocrystals prepared by high-power ultrasound had similar dissolution profiles, as differences in their dissolution were not statistically significant. On the other hand, cocrystals prepared by HPH had a higher dissolution at 600 and 900 bar compared to 300 bar from 3–10 min, however they were similar at 20 min of dissolution. The highest dissolution was seen from cocrystals prepared by 600 and 900 bar. Cocrystals prepared by HPH at 600 bar had a higher dissolution of SMT, which is unexpected as the smaller-size cocrystals generated at 900 bar were expected to have enhanced dissolution. A closer look at the DLS measurements showed that cocrystals prepared at 900 bar had a higher hydrodynamic radius compared to those prepared at 600 bar (Appendix A). According to the Stokes–Einstein equation, the diffusion coefficient is inversely proportional to the hydrodynamic radius [59]. Therefore, it can be deduced that the lower diffusion coefficient for the cocrystals prepared at 900 bar is due to the agglomerates, which could explain the difference in the dissolution rate of nano-cocrystals prepared at 600 and 900 bar (Figure 5 and Appendix A). Moreover, despite the size of nano-particles being considered as the main physicochemical property affecting solubility, other parameters like crystallinity, surface morphology, and surface area should also be taken into account, because the surface bond strength and spatial arrangements and the presence of adatoms have an influence on the dissolution of nano-particles [60].

## 4. Conclusions

Nano-sized multi-drug cocrystals of 4-ASA and SMT were successfully prepared by HPH, while the high-power ultrasound crystallization method only yielded cocrystals with a mean size in the micro-size range. Utilizing HPH with 900 bar pressure and 5 homogenization cycles resulted in a smaller size with a narrow size distribution, while the cocrystals prepared by high-power ultrasound and HPH with lower homogenization pressures had a wide particle size distribution. The morphology of the cocrystals depended on the preparation method. High-power ultrasound resulted in cocrystals with various habits and morphologies, while the HPH method resulted in needle-shaped cocrystals, as observed by TEM. All cocrystals formed by both methods were found to be of the stable polymorphic form I, Triclinic, P-1. Nano-sizing by HPH significantly improved the dissolution rate compared to micro-size cocrystals and even more compared to pure APIs. Nano-cocrystals have been found to be stable for 6 months without ac hange in morphology or size distribution after storage at room temperature in a desiccator. Further studies are needed to examine the effect of other stabilizers on nano-sized cocrystals and enhance process optimization.

## Figures and Tables

**Figure 1 pharmaceutics-13-00277-f001:**
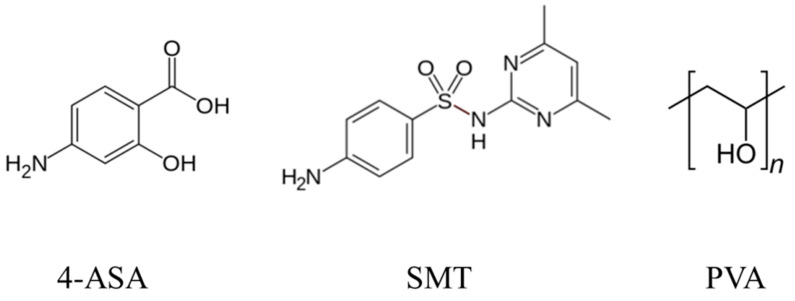
Active pharmaceutical ingredients (APIs) and the stabilizer used to prepare the cocrystals. 4-ASA: 4-aminosalysilic acid; SMT: sulfamethazine; PVA: polyvinyl alcohol.

**Figure 2 pharmaceutics-13-00277-f002:**
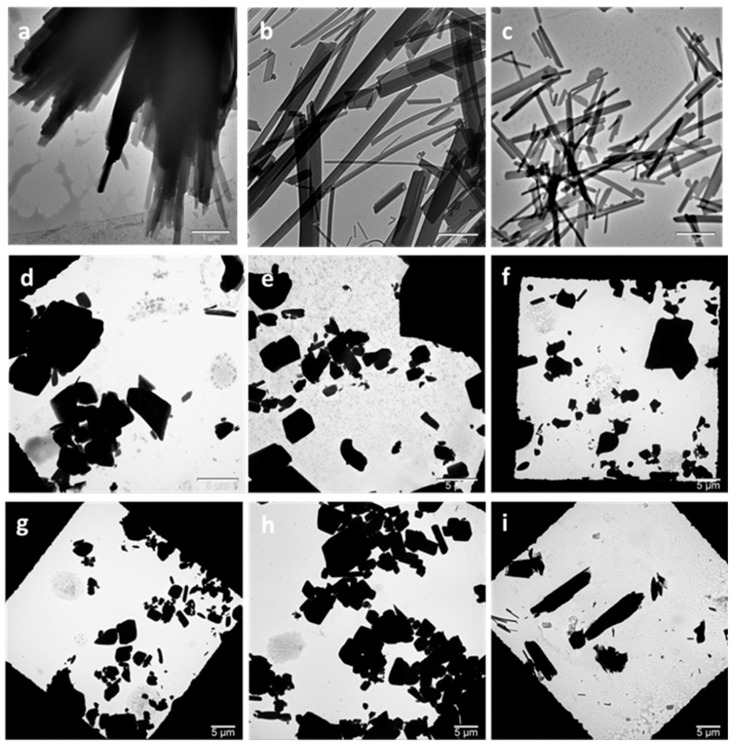
Transmission electron microscopy (TEM) images of HPH- and high-power ultrasound-prepared cocrystals. (**a**) HPH 300 bar 5c, (**b**) HPH 600 bar 5c, (**c**) HPH 900 bar 5c, (**d**) HPU 50% 10 min, (**e**) HPU 70% 10 min, (**f**) HPU 70% 20 min, (**g**) HPU 70% 30 min, (**h**) HPU 90% 10 min, and (**i**) control. HPH: high pressure-homogenization; HPU: high-power ultrasound; c: number of HPH cycles; percentage refers to HPU amplitude.

**Figure 3 pharmaceutics-13-00277-f003:**
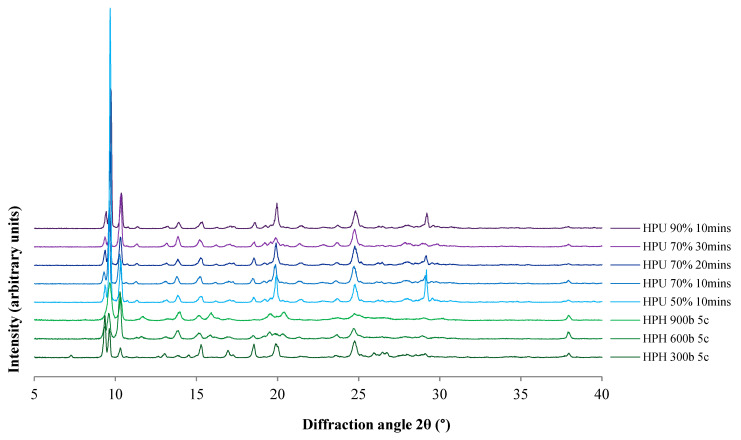
Powder X-ray diffraction (PXRD) patterns of the cocrystals prepared by HPH and high-power ultrasound. HPH: high pressure-homogenization; HPU: high-power ultrasound; c: number of HPH cycles; percentage refers to HPU amplitude.

**Figure 4 pharmaceutics-13-00277-f004:**
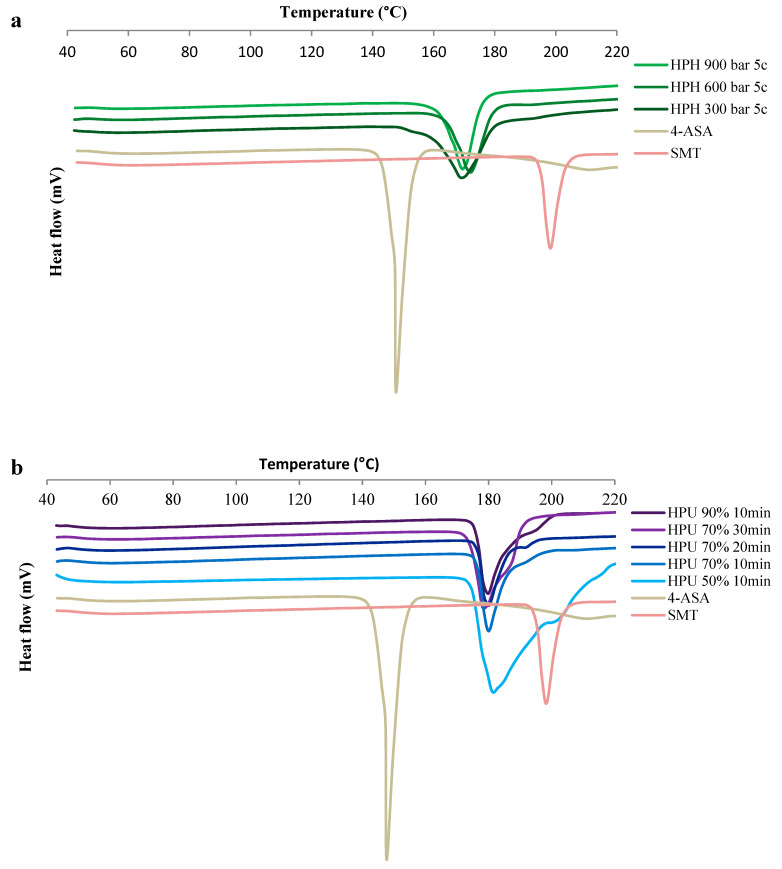
Differential scanning calorimetry (DSC) thermograms of the cocrystals. (**a**) Cocrystals prepared by HPH and (**b**) cocrystals prepared by HPU. HPH: high pressure-homogenization; HPU: high-power ultrasound; c: number of HPH cycles; percentage refers to HPU amplitude.

**Figure 5 pharmaceutics-13-00277-f005:**
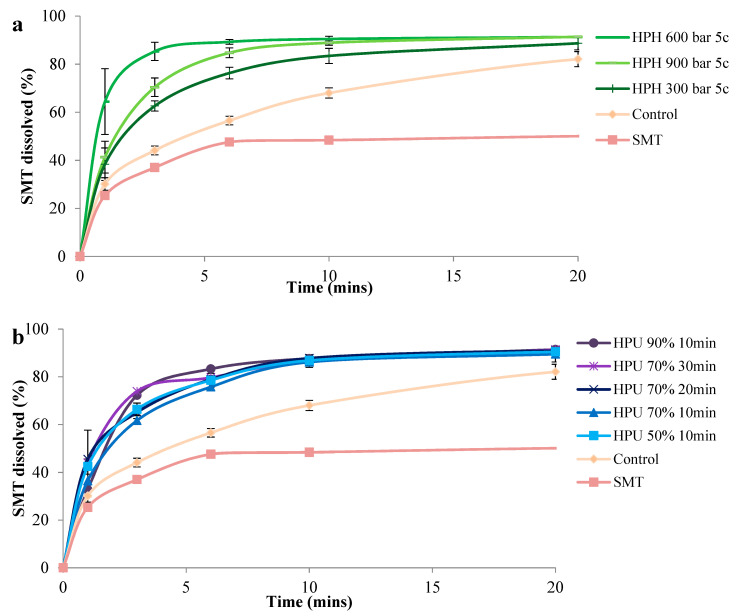
SMT dissolution profiles of the cocrystals. (**a**) Cocrystals prepared by HPH and (**b**) cocrystals prepared by HPU. HPH: high pressure-homogenization; HPU: high-power ultrasound; c: number of HPH cycles; percentage refers to HPU amplitude.

**Table 1 pharmaceutics-13-00277-t001:** Cocrystal size as measured from TEM images.

Sample	Dimension	Size from TEM (nm)			
		Mean ± SD	D10	D50	D90
HPH 900b 5c	Width	80.90 ± 24.68	54.20	80.80	117.66
	Length	1158.69 ± 859.25	318.77	893.86	2716.94
HPH 600b 5c	Width	118.61 ± 35.02	79.10	119.06	159.26
	Length	2528.91 ± 2104.28	625.91	2008.74	5131.60
HPH 300b 5c	Width	142.00 ± 32.09	109.87	131.80	194.63
	Length	3137.88 ± 1467.43	2128.29	2543.27	4310.08
HPU 90% 10 min		2590.9 ± 2311.35	1096.35	1931.25	3767.95
HPU 70% 30 min		2177.72 ± 1212.25	985.00	1924.13	3813.13
HPU 70% 20 min		2353.09 ± 1668.87	1075.60	1954.25	3843.13
HPU 70% 10 min		1854.02 ± 1106.04	597.61	1761.55	3006.02
HPU 50% 10 min		4472.79 ± 3718.24	1665.07	2937.13	11,129.10
Control		4950.86 ± 2989.08	2257.83	4260.75	8252.65

HPH: high pressure-homogenization; HPU: high-power ultrasound; c: number of HPH cycles; percentage refers to HPU amplitude.

## Data Availability

Data is contained within the article or Appendix A.

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
