# Peer review of "Synthesis and Characterization of Nano-Sized 4-Aminosalicylic Acid–Sulfamethazine Cocrystals"

_pharmaceutics, 2021, doi:10.3390/pharmaceutics13020277_

Round 1
Reviewer 1 Report
Dear Editor,
This contribution by A. Salem and coworkers reports on the synthesis of a drug-drug cocrystal (4-aminosalycilic acid/sulfamethazine) via high-pressure homogenization (HPH) and high-power ultrasound (HPU), and their physical characterization by TEM, XRPD, DSC and DLS. Also, authors performed powder dissolution experiments to assess differences between these solids.
Notwithstanding the manuscript is correctly presented, I face questions over their scientific goals because they are limited only to practical implications and routine findings. Allow me to elaborate further this previous comment. The cocrystal studied is not novel, it was previously reported in an extensive and comprehensive investigation by A. M. Healy and coworkers in 2015 (See Ref. 22). The current authors have recently published an interesting study on polymorphic forms of this cocrystal (See Ref. 38). The present work looks like a simple extension of those findings.
In the introduction, authors did not convincingly explain what the benefits are of combining nanosizing and cocrystals approaches. What is the expected advantage of downsizing a cocrystal versus the pristine APIs?
In the results and discussion section, it is implied that the main objective of generating “nano-cocrystals” was reached by using HPH. However, by looking at TEM images (particularly Figure 2c) it is clear that micrometer size cocrystals were mostly obtained with a wide multimodal size distribution by DLS. In short, this methodology did not afford a homogeneous distribution of “nano-cocrystals”.
Finally, authors presented powder dissolution studies of these cocrystalline solids to demonstrate a functional advantage over a control sample of the cocrystal (not micronized). Dissolution profiles were presented as % SMT dissolved (SMT= sulfamethazine), however, it is not clear how to account for the contribution of the other component (4-aminosalicylic acid is also a chromophore) if quantification was performed only by UV/Vis spectroscopy. As reported in reference 22, an accurate quantification of both components is only achievable by HPLC analysis.
In summary, I do not recommend this manuscript for publication in Pharmaceutics.
Reviewer 2 Report
The presented paper is devoted to the very actual and important field of modern pharmaceutics developing the methods of drug substances modification and formation of new nanosized crystals and co-crystals. Such experimental approach is aimed to enhance the solubility and bioavailability of active pharmaceutical ingredients (APIs). Aiming to develop pharmaceutical combinational therapy or fixed-dose medication, pharmaceutical drug-drug co-crystals offers an convenient approach when a specific disease is treated by a combination of several APIs
The two methods were used in the paper - high-pressure homogenization (HPH) and high-power ultrasound (HPU) in order to prepare nanosized co-crystals of 4-aminosalysilic acid and sulfamethazine. The results obtained in the paper establish differences between the two methods in terms of co-crystal size, morphology, polymorphic form and dissolution rate enhancement and other physicochemical properties. The morphology of the co-crystals was depended on the preparation method. Co-crystals obtained by HPU resulted in co-crystals with various habits and morphologies, while HPH method resulted in needle shaped co-crystals according to TEM-data. The co-crystals formed by both methods were found to be in the stable polymorphic form - Triclinic, P-1. HPH-method significantly increased the dissolution rate compared to micro-sized co-crystals obtained by HPU technique. Nano co-crystals have been found to be stable during 6 months storage at room temperature in desiccator without any changes in morphology or size distribution.
Thus, the presented paper contained the new experimental results on physico-chemical properties of nanosized crystals and co-crystals of two highly used drug substances and establish the possibilities of HPH and HPU methods applied to the production of drug substances nanoforms. Further studies directed on the optimization of nanosized crystals and co-crystals formation process will be of practical interest for modern pharmaceutics and nanomedicine.
The paper was written in clear and well understandable form using good scientific English and can be published in present form after minor editor corrections.
Reviewer 3 Report
The combination of cocrystallization and particle size reduction can amplify the dissolution enhancement caused by either methods, and therefore this manuscript presents a very interesting idea. However there are some points that need to be considered by the authors before the manuscript can be accepted for publication, which are listed below.
- Line 192, filtering the samples through a 0.200 micrometer filter when the nanocrystal Z-average diameter is as low as 80 nanometers in some cases, is problematic. A smaller filter should be used, or an alternative method.
- Methods section 2.7 statistical analysis. A description of the statistical methodology used in this work should be provided here, not just a reference to the software used.
- page 7, lines 228-236. Why is line broadening solely attributed to reduction of crystallinity, when its sources can (and most likely are) crystallite size reduction and microstrain?
- Figure 5. All dissolution curves level off at around 85% SMT dissolved. Is that an effect of non sink conditions or some polymorpic transformation of SMT that changes its solubility?
Round 2
Reviewer 1 Report
The revised version of the contribution by A. Salem and coworkers presents some improvements. I am now inclined to recommend this manunscript for publication in Pharmaceutics but the following comments must be addressed to by the authors, particularly the first two issues are of relevance:
- According to authors, dissolution experiments are key to evaluate differences between nano-cocrystals and controls. Therefore an accurate method to quantify SMT is of utmost importance. Please include relevant information and details (spectra of starting materials, calibration curves, statistics, etc.) to demonstrate that UV/vis spectroscopy is indeed selective to quantify SMT in the presence of 4-ASA as it is found in the dissolution samples.
- As far as one can guess in Figure S3 (which is not a good example of presenting clearly meaningful data) DLS experiments showed broad multimodal size distributions mostly in the micrometer range. Authors should acknowledge that information in the discussion section (p.10 lines 267-273) and conclusions. To give a fair assessment of particle size distributions, data should be given as D10, D50 and D90 (see for instance CrystEngComm, 2020,22, 2304-2314).
- I believe that more citations must be included to sustain that “Nano-sized crystals have gained substantial attention” (p. 3, lines 90-91). Only one reference is cited and I just found that is incomplete (see ref. 24).
- Cocrystal is now commonly used without hyphen (not Co-crystal).
Round 3
Reviewer 1 Report
I am glad to read this revised version. I acknowledge that authors have improved the manuscript and responded properly to all inquiries. I have no further comments and recommend this contribution for publication.